# Chemical modulation of transcriptionally enriched signaling pathways to optimize the conversion of fibroblasts into neurons

Joseph Herdy[1], Simon Schafer[1], Yongsung Kim[1], Zoya Ansari[1], Dina Zangwill[1], Manching Ku[2], Apua Paquola[3], Hyungjun Lee[1], Jerome Mertens[1,4]*, Fred H Gage[1]*

[1]Laboratory of Genetics, The Salk Institute for Biological Studies, La Jolla, United States; [2]University Hospital Freiberg, University of Freiberg, Breisgau, Germany; [3]Lieber Institute for Brain Development, Baltimore, United States; [4]Institute of Molecular Biology, CMBI, Leopold-Franzens-University Innsbruck, Innsbruck, Austria

**Abstract** Direct conversion of human somatic fibroblasts into induced neurons (iNs) allows for the generation of functional neurons while bypassing any stem cell intermediary stages. Although iN technology has an enormous potential for modeling age-related diseases, as well as therapeutic approaches, the technology faces limitations due to variable conversion efficiencies and a lack of thorough understanding of the signaling pathways directing iN conversion. Here, we introduce a new all-in-one inducible lentiviral system that simplifies fibroblast transgenesis for the two pioneer transcription factors, Ngn2 and Ascl1, and markedly improves iN yields. Further, our timeline RNA-Seq data across the course of conversion has identified signaling pathways that become transcriptionally enriched during iN conversion. Small molecular modulators were identified for four signaling pathways that reliably increase the yield of iNs. Taken together, these advances provide an improved toolkit for iN technology and new insight into the mechanisms influencing direct iN conversion.
DOI: https://doi.org/10.7554/eLife.41356.001

*For correspondence:
jmertens@salk.edu (JM);
gage@salk.edu (FHG)

## Introduction

Human somatic cells such as skin fibroblasts can be directly converted into cultures of functional induced neurons (iNs) by the overexpression of pro-neuronal transcription factors (*Pang et al., 2011*; *Chambers and Studer, 2011*). As opposed to induced pluripotent stem cell (iPSC) reprogramming and differentiation, direct iN conversion bypasses the pluripotent stage as well as any other stem cell-like stages and preserves epigenetic information of their donor's age, making it a particularly valuable tool to study aging and aging-related disorders (*Mertens et al., 2015*; *Victor et al., 2018*; *Tang et al., 2017*). iN technology has also shown promise in vivo as a strategy to replace damaged cells following brain injury by direct conversion of non-neuronal cell types into neurons directly within the nervous system (*Karow et al., 2012*; *Heinrich et al., 2010*). Using combinations of pro-neuronal and region-/subtype-specific transcription factors, a variety of neuronal subtypes has been produced via direct conversion (*Caiazzo et al., 2011*; *Son et al., 2011*; *Victor et al., 2014*; *Vadodaria et al., 2016*; *Tsunemoto et al., 2018*).

Because iN conversion lacks a proliferating stem cell intermediate, the iN numbers obtained are largely dependent on conversion efficiency; therefore, great efforts have been undertaken to improve iN process yields. The identification and the combination of Ngn2 with Ascl1 as two pro-neuronal pioneer transcription factors that can induce neuronal identity in non-neuronal cells have been key features in the advancement of this technology (*Ladewig et al., 2012*; *Liu et al., 2013*;

*Mertens et al., 2015*; *Wapinski et al., 2013*; *Smith et al., 2016*). The efficient delivery of Ngn2/Ascl1 into fibroblasts, and their robust, controllable transgene expression, left room for improvement and, because variations in the transduction and conversion efficiencies from different donors are common limitations, we have developed a new all-in-one lentiviral system for inducible Ngn2/Ascl1 expression called UNA.

Chemical modulation of several cellular signaling pathways has been shown to improve iN conversion and thus enable iN technology as a legitimate alternative to iPSC differentiation. Inhibition of TGF-b/SMAD signaling via blockade of AKT kinases, inhibition of GSK-3b signaling (*Ladewig et al., 2012*), adenylyl cyclase activation (*Liu et al., 2013*; *Ladewig et al., 2012*; *Gascón et al., 2016*), inhibition of REST (*Masserdotti et al., 2015*; *Xue et al., 2013*), induction of canonical Bcl-2 signaling, and inhibition of lipid oxidation pathways via ferroptosis inhibition (*Gascón et al., 2016*) have all been shown to greatly promote iN conversion. However, because these known signaling pathway modulators have either been adopted from iPSC differentiation (*Chambers et al., 2009*) or are the result of educated guessing or simply of trial-and-error experiments, no systematic efforts have been made to utilize broad and unbiased datasets to identify the pathways and corresponding modulators that orchestrate iN conversion. We sought to optimize the iN conversion media composition with small molecular pathway modulators to reach efficient iN conversion even in suboptimal human fibroblast lines. To that end, we have performed timeline RNAseq transcriptome analysis over the time course of direct iN conversion and have discovered four pathways that are instrumental in iN conversion. Based on these pathways, we identified four small molecules that could be combined into an improved iN conversion medium, ZPAK, which reliably boosted iN conversion of young and old fibroblasts into epigenetically age-equivalent iNs.

## Results

### An optimized all-in-one viral system simplifies fibroblast transduction and increases conversion efficiency

Lentiviral delivery of pro-neuronal transcription factors is the most widely used technique for direct iN conversion due to its outstanding efficiency and relative ease to use (*Parr-Brownlie et al., 2015*). Successful strategies involve the use of constitutive or inducible expression of either only a single pioneer factor, such as Ngn2 or Ascl1 (*Liu et al., 2013*; *Smith et al., 2016*), a combination of a pioneer factor with secondary factors (*Pang et al., 2011*; *Vierbuchen et al., 2010*), a combination of a pioneer factor with subtype-specifying transcription factors (*Caiazzo et al., 2011*; *Liu et al., 2016*; *Vadodaria et al., 2016*); (*Tsunemoto et al., 2018*), or the use of two coupled pioneer factors (*Ladewig et al., 2012*; *Mertens et al., 2015*). Typically, these factors are distributed across several lentiviral vectors; however, given that lentiviral transduction is not 100% efficient and because an individual fibroblast must be transduced with two or more viruses to survive chemical selection and successfully reprogram, this strategy left room for improvement. To eliminate the need for co-infection, we combined the tetOn system cassette consisting of the rtTA$_{Adv.}$ [Clonech] driven by the UbC promoter, the iN cassette consisting of Ngn2:2A:Ascl1 under control of the TRE$_{tight}$ promoter [Clontech], and a puromycin-resistance gene driven by the PGK promoter to yield the UNA construct (*Figure 1A*, *Figure 1—figure supplement 1*). To test the efficiency of UNA compared to our conventional two-vector system (EtO +N2A), we selected fibroblasts from three individual donors that had not yielded optimal iN conversion efficiencies in the past (*Figure 1B* and *Figure 1—source data 1*). Dermal fibroblasts were transduced with similar titers of either EtO +N2A or UNA, selected by puromycin or puromycin/G418, respectively, and converted to iNs using our previously described conversion medium (NC) containing noggin as well as the small molecules CHIR-99021 (GSK3ß inhibitor), SB-431542, LDN-193189, A-83–01 (ALK inhibitors), forskolin, and db-cAMP (cAMP increase) (*Mertens et al., 2015*; *Ladewig et al., 2012*). Following three weeks of conversion, cells were live-stained for the neural surface marker PSA-NCAM, which stains reprogrammed iNs but not fibroblasts (*Figure 1C*). Flow cytometry revealed that UNA-derived iNs exhibited significantly more PSA-NCAM-positive cells than E + N2A, boosting efficiencies by up to 90–100% for the two suboptimal fibroblast lines, but also increasing efficiencies of the lines that already converted well by 30 ± 8% (*Figure 1D–E*). In addition, immunocytochemical co-staining for the neuronal markers ßIII-tubulin and NeuN revealed that UNA iNs were on average twice as likely to be positive for NeuN or ßIII-

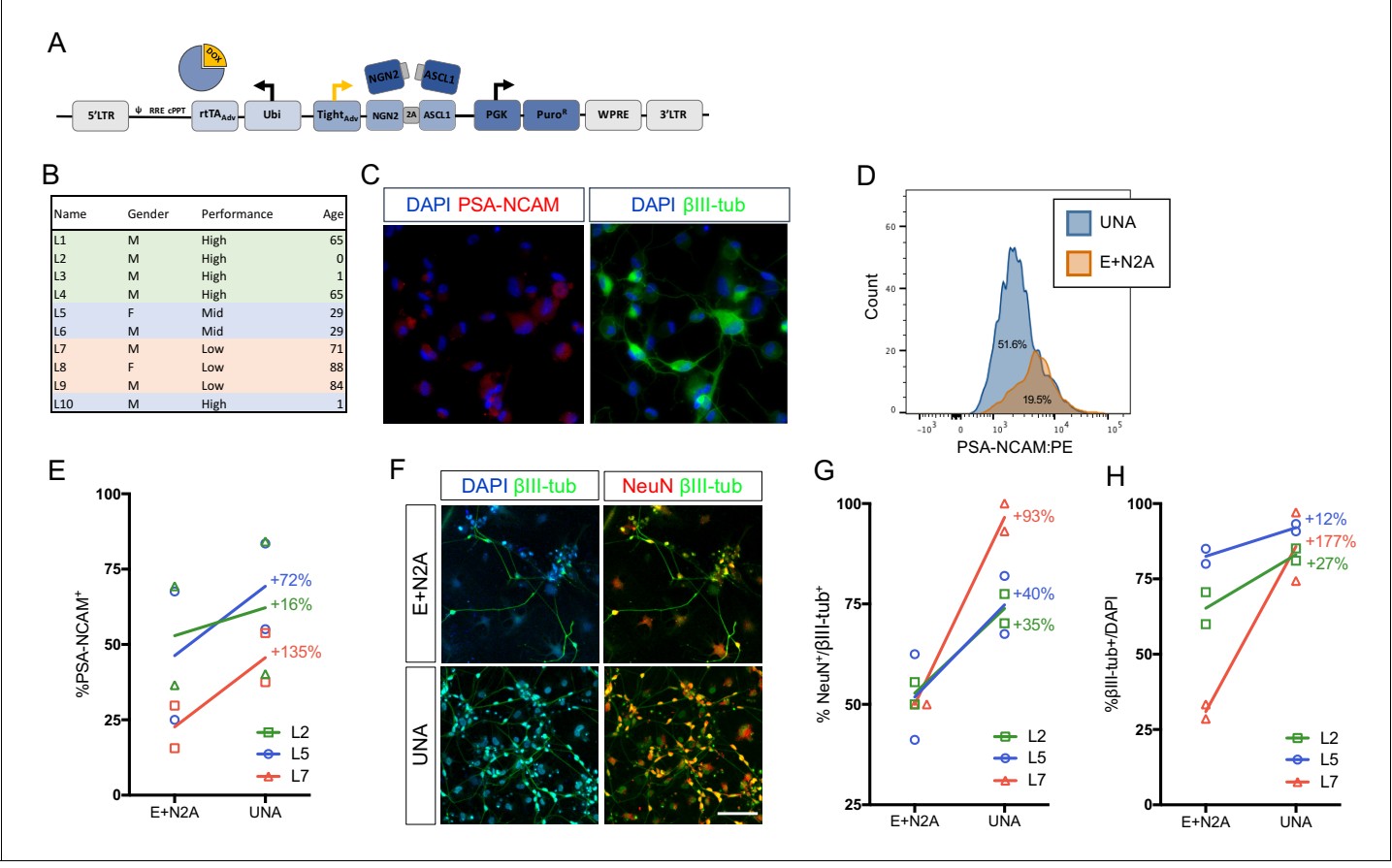

**Figure 1.** An optimized all-in-one viral system simplifies fibroblast transduction and increases conversion efficiency. (A) Schematic of all-in-one lentiviral system for inducible overexpression of N2A for iN conversion. (B) Cell lines of varying conversion efficiencies used for comparison (C) Immunocytochemical analysis of iNs following three weeks of conversion, stained with βIII-tubulin, PSA-NCAM, and DAPI. The scale bar represents 20 μm. (D) Flow cytometry histogram plots of PSA-NCAM:PE-stained iNs following 3 weeks of conversion with UNA (blue) or E:N2A (Orange) lentiviral systems. (E) Comparison of % PSA-NCAM:PE + cells from three three lines (L2,L5,L7) reprogrammed with E + N2A or UNA (3 biological and two technical replicates). (F) Immunocytochemical analysis of E:N2A or UNA iNs following three 3 weeks of conversion, stained with βIII-tubulin, NeuN, and DAPI. Scale bar represents 100 μm. (G-H) Quantification of immunocytochemical staining in F. (3 biological and two technical replicates).

DOI: https://doi.org/10.7554/eLife.41356.002

The following source data and figure supplement are available for figure 1:

**Source data 1.** Human fibroblasts used in this study.

DOI: https://doi.org/10.7554/eLife.41356.004

**Figure supplement 1.** Heatmap of FPKM normalized counts of Neurog2 and Ascl1.

DOI: https://doi.org/10.7554/eLife.41356.003

tubulin (*Figure 1F–H*). Based on these experiments, we reasoned that the use of the single all-in-one lentiviral vector UNA was notably easier and less prone to experimental error, as well as significantly more efficient in generating larger numbers of mature iNs compared to a multiple vector strategy.

## Time based RNAseq identifies signaling pathways directing iN conversion

As pathways influential in controlling the direct conversion process continue to be found (*Masserdotti et al., 2015*; *Smith et al., 2016*; *Treutlein et al., 2016*; *Xue et al., 2013*; *Liu et al., 2013*; *Ladewig et al., 2012*; *Gascón et al., 2016*), we decided to investigate the transcriptional dynamics during reprogramming to identify the pathways that orchestrate iN conversion. We gathered RNAseq data from bulk fibroblasts as well as cells undergoing conversion for 1, 2, 3, 6, 18, and 24 days (Line 10, *Figure 2A*). Using the Ingenuity Pathway Analysis software tool (IPA; Qiagen Inc),

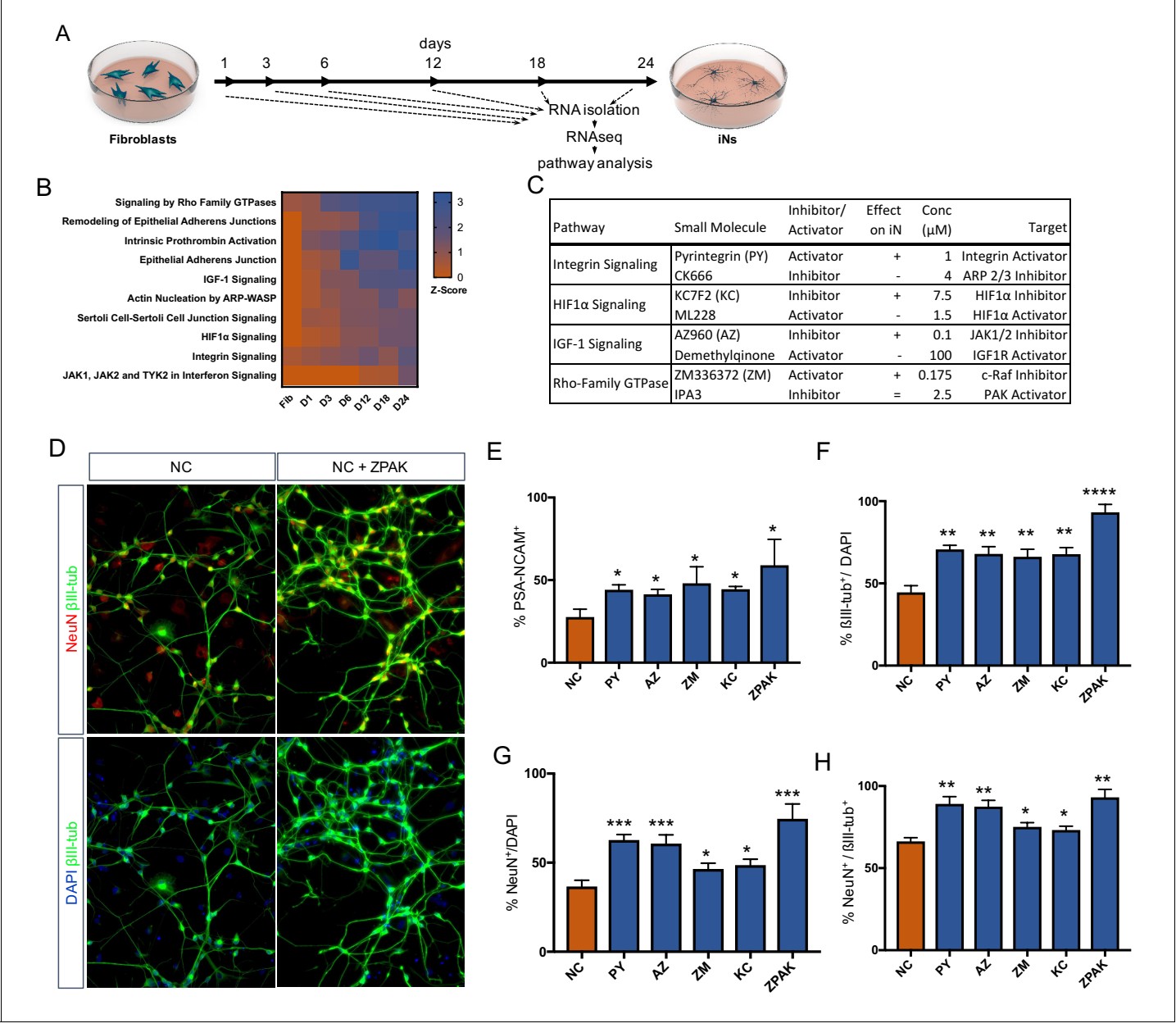

**Figure 2.** Time-based RNAseq identifies signaling pathways influential during iN conversion. (A) Schematic for timeline of RNA isolation during neuronal reprogramming of L10 (B) Activation Z-Score of the indicated signaling pathway as called by IPA Comparison Analysis during neuronal reprogramming. Z-Scores represent a statistical measure of the match between the expected relationship direction and observed gene expression of a given pathway, with z-scores > 2 considered significant. (C) Description of small molecules selected to inhibit or activate branches of pathways identified in (B) Effect on iN conversion is based on increased % PSA-NCAM + cells by flow cytometry, with + indicating increased PSA-NCAM %, - reduced PSA-NCAM %, and = no effect on PSA-NCAM %. (D) Immunocytochemical analysis of neural markers in NC or NC+ZPAK iNs (L1, L5, L9) following three weeks of conversion, stained with βIII-tubulin, NeuN, and DAPI. Scale bar represents 100 μm. Representative image from L1. (E) flow cytometry quantification of %PSA-NCAM + cells in iNs (L1, L5, L9) converted in NC and NC+ZPAK supplements. (F-H) Quantification of immunocytochemical staining in (D). Results are shown as mean ± SEM. *p<0.05; **p<0.01; ***p<0.001; ****p<0.0001, n = 3 biological replicates. Significance values were calculated by t test.

DOI: https://doi.org/10.7554/eLife.41356.005

The following source data and figure supplements are available for figure 2:

**Source data 1.** Ingenuity pathway analysis of direct fibroblast to neuron conversion.
DOI: https://doi.org/10.7554/eLife.41356.010
**Source data 2.** Expanded small molecule information.

*Figure 2 continued on next page*

*Figure 2 continued*

DOI: https://doi.org/10.7554/eLife.41356.007

**Figure supplement 1.** Effect of screened small molecules on %PSA-NCAM yield of iNs.

DOI: https://doi.org/10.7554/eLife.41356.006

**Figure supplement 2.** % yield of PSA-NCAM+ iNs from all 10 lines used in this study.

DOI: https://doi.org/10.7554/eLife.41356.008

**Figure supplement 3.** Concentration optimization of ZPAK cocktail.

DOI: https://doi.org/10.7554/eLife.41356.009

we identified more than 500 pathways called to be significantly transcriptionally enriched or depleted ($-2 >$ Z Score$>2$) as conversion proceeded (*Figure 2—source data 1*); of these, we selected 10 of the most enriched pathways for further experimental testing (*Figure 2—source data 2*). Based on the regulated genes and overlap of the called pathways, we selected one small molecule activator and one inhibitor for each of the 10 pathways, with the strategy of initiating diametrically opposing regulation of a given pathway. Each of the 20 compounds had been previously been reported to be effective in tissue culture (*Figure 2—source data 2*). Preference was given to molecules that targeted an intersection of two or more called pathways rather than specifically targeting the canonical signaling cascade of one pathway. When screening for PSA-NCAM-positive cells by flow cytometry following 21 days in NC plus the respective compound, four compounds were found to significantly ($p<0.05$) increase the frequency of PSA-NCAM-positive iNs (*Figure 2C–E*, *Figure 2—source data 2* and *Figure 2—figure supplement 1*) Interestingly, the compound that acted in the opposite direction decreased iN conversion efficiency (*Figure 2C–D*, *Figure 2—source data 2*, and *Figure 2—figure supplement 1*). These four conversion booster compounds were Pyrintegrin (PY; Integrin activator), AZ960 (AZ; Jak2 inhibitor), ZM336372, (ZM; Raf-1 activator), and KC7F2 (KC; HIF1α inhibitor) (*Figure 2C*). The iNs derived in NC plus any of these four compounds resulted in a significant ($p<0.01$) increase in the number of NeuN- and ßIII-tubulin-positive cells compared to those derived in NC medium alone (*Figure 2F–H*, *Figure 2—figure supplement 2*). Importantly, the combination of all four compounds (ZPAK) resulted in an even higher iN yield than any of the compounds individually (*Figure 2F–H*, *Figure 2—figure supplement 3*). This combination of all four molecules with the NC medium will henceforth be referred to as ZPAK.

## ZPAK induces a defined neuronal transcription that more closely relates to the adult brain transcriptome than NC alone

To investigate the changes induced on the transcriptional level by the ZPAK cocktail, we again performed time series RNAseq from fibroblasts from three donors that were converted to iNs in NC or NC+ZPAK for 5, 10, 15, and 20 days (*Figure 3A*; three biological replicates from four time points, 24 samples total). Comparing FPKM normalized gene counts for the conversion process for both conditions, we detected that the transcriptomes induced by NC or NC+ZPAK were highly correlated, with a Pearson correlation coefficient of $\geq 0.85$ for all sampled time points (*Figure 3C*), indicating broad transcriptional similarities between the iN process in NC and NC+ZPAK. Comparing the top 100 upregulated and top 100 downregulated genes for the conversion process for both conditions, we detected an 80% and 93% overlap, respectively (*Figure 3B*). Consistently, glutamatergic neuron-specific genes Unc-13 homolog A (*UNC13A*), AMPA receptor auxiliary protein 2 (*CNIH2*), NMDA1 (*GRIN1*), GluK5 (*GRIK5*), GluK2 (*GRIK2*) and vGLUT1 (*SLC17A7*), and the GABAergic neuron-specific genes phospholipase C like 2 (*PLCL2*), dopamine receptor D2 (*DRD2*), cannabinoid receptor 1 (*CNR1*) and glutamate decarboxylase 1 (*GAD1*) were found to be generally expressed at similar levels in NC and NC+ZPAK at 20 days of conversion (*Figure 3D*, *Figure 3—figure supplement 1*). Markers of dopaminergic, serotonergic, and cholinergic lineages were not consistent in expression and were not commonly observed in D20 iNs (*Figure 3D*, *Figure 3—figure supplement 2*). We next performed differential expression analysis between all NC and NC+ZPAK time points, revealing 143 genes that were significantly differentially expressed (*Figure 3—source data 1*, padj $< 0.05$). Hierarchical clustering demonstrated a clear separation of NC and NC+ZPAK groups except for two NK samples at the earliest point of conversion (*Figure 3E*). These significant transcriptional differences point to a more defined neuronal transcriptome initiated in NC+ZPAK compared to NC. Interestingly, we observed consistently increased expression of neuron-specific genes

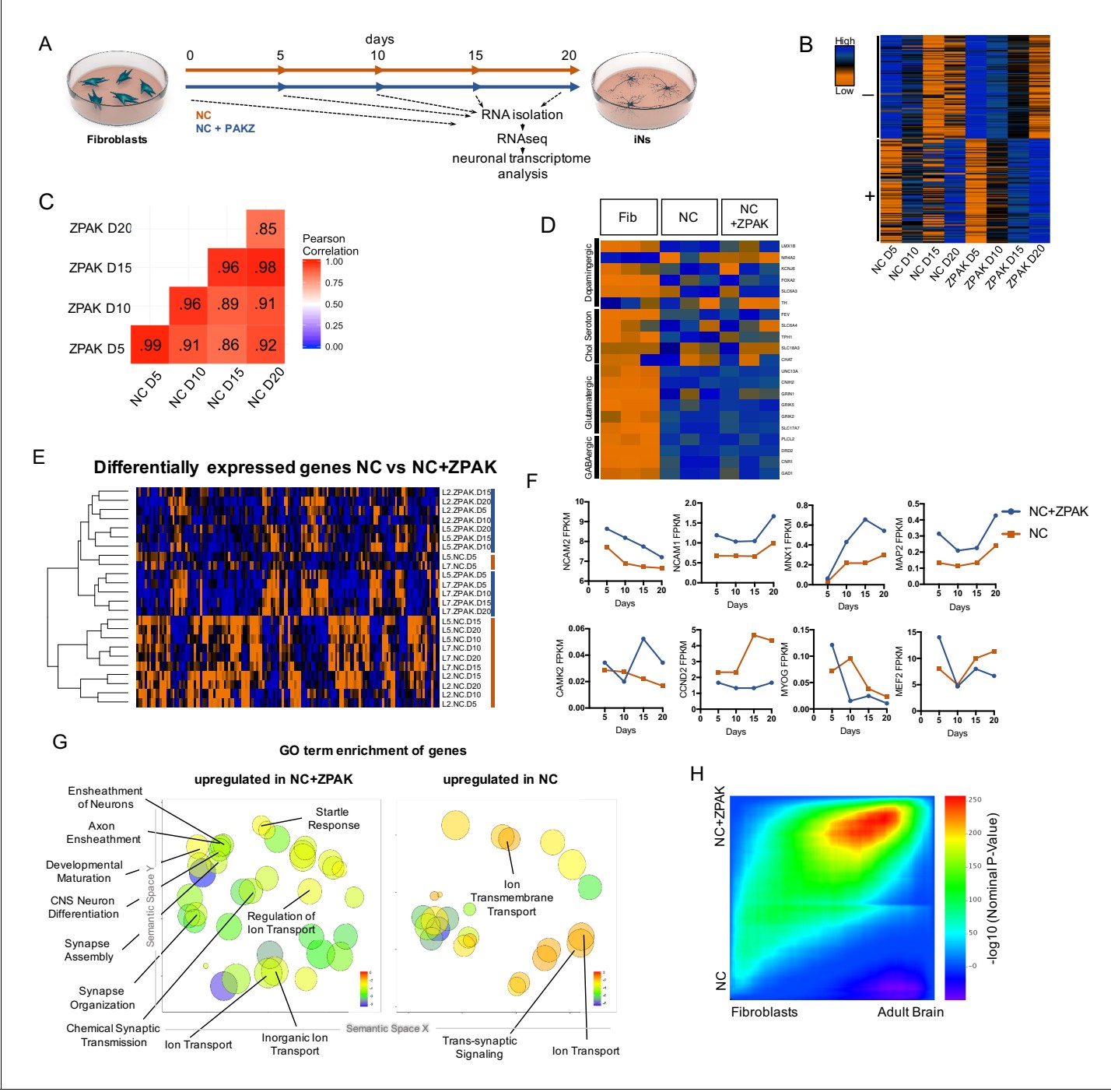

**Figure 3.** ZPAK induces a more defined neuronal transcription that more closely relates to the adult brain transcriptome. (A) Schematic for timeline RNA isolation during neuronal reprogramming of L2, L5, and L7 at fibroblast and 5, 10, 15, and 20 days of conversion (B) Heatmap of FPKM normalized counts of the top 100 correlating (+) and top 100 inverse correlating (-) genes (FPKM ≥ 1) for day 5, 10, 15 and 20 of neuronal reprogramming with NC or NC+ZPAK. Orange = low expression, blue = high expression (C) Correlate R values of FPKM normalized counts from 25,610 genes during neuronal reprogramming with NC or NC+ZPAK cocktail. (D) Heatmap showing relative expression of the glutamatergic neuron-specific genes Unc-13 homolog A (UNC13A), AMPA receptor auxiliary protein 2 (CNIH2),NMDA1 (GRIN1), GluK5 (GRIK5), GluK2 (GRIK2) and vGLUT1 (SLC17A7), GABAergic neuron-specific genes phospholipase C like 2 (PLCL2), dopamine receptor D2 (DRD2), cannabinoid receptor 1 (CNR1) and glutamate decarboxylase 1 (GAD1), Serotonergic neuron-specific genes ETS transcription factor (FEV), serotonin transporter 1 (SLC6A4), and tryptophan hydroxylase (TPH1), Dopaminergic neuron-specific genes tyrosine hydroxylase (TH), dopamine transporter 1 (SLC6A3), forkhead box A2 (FOXA2), potassium voltage-gated channel subfamily J member 6 (KCNJ6), nuclear receptor subfamily four group A member 2(NR4A2), and LIM homeobox transcription factor one beta (LMX1B), and cholinergic neuron-specific genes vesicular acetylcholine transporter (SLC18A3), and choline O-acetyltransferase (ChAT); normalized by row. (E)

*Figure 3 continued on next page*

*Figure 3 continued*

Heatmap of 143 significantly (*p-adj* < 0.05) differentially expressed genes between NC and NC+ZPAK at 5, 10, 15 and 20 days of reprogramming (*n* = 3). (F) FPKM normalized counts of five representative neuron specification genes - Neural Cell Adhesion Molecule 2 (NCAM2), Neural Cell Adhesion Molecule 1 (NCAM1), Motor Neuron and Pancreas Homeobox 1 (MNX1), Microtubule-associated Protein 2 (MAP2), Calcium/Calmodulin Dependent Protein Kinase II (CAMK2) - and three representative fibroblast-to-iN limiting genes - Cyclin D2 (CCND2), Myogenin (MYOG), and Myocyte enhancer factor-2 (MEF2) - over time during fibroblast-to-iN conversion with NC (orange) or NC+ZPAK (blue). (G) Gene ontology (GO) enrichment analysis of genes upregulated in NC or NC+ZPAK (log$_2$FC > 1). Results are shown as REVIGO (*Supek et al., 2011*) scatterplots in which similar GO terms are grouped in arbitrary two-dimensional space based on semantic similarity. Each circle indicates a specific GO term and circle sizes are indicative of how many genes are included in each term, where larger circles indicate greater numbers of genes that are included in that GO term. Colors indicate the level of significance of enrichment of the GO term by log$_{10}$ p-value. (H) Rank-rank hypergeometric overlap (RRHO) map (*Plaisier et al., 2010*) comparing the gene expression differences between NC and NC+ZPAK to expression differences between matched fibroblasts and adult human brain (Allen Brainspan). Each pixel represents overlap between NC or NC+ZPAK to fibroblast or adult brain transcriptome, color-coded according to the –log$_{10}$p value of a hypergeometric test (step size = 100). On the map, the extent of shared genes upregulated in NC+ZPAK and adult brain is displayed in the top right corner, whereas the shared genes upregulated in NC and fibroblasts are displayed in the bottom left corner (see schematic in *Figure 3—figure supplement 6*).

DOI: https://doi.org/10.7554/eLife.41356.011

The following source data and figure supplements are available for figure 3:

**Source data 1.** Significantly differentially expressed genes between NC and NC+ZPAK.

DOI: https://doi.org/10.7554/eLife.41356.018

**Figure supplement 1.** RRHO schematic adapted from RRHO User Guide (*Plaisier et al., 2010*).

DOI: https://doi.org/10.7554/eLife.41356.012

**Figure supplement 2.** Immunocytochemical quantification of neuron subtype markers TH (Dopaminergic), vGlut1 (Glutamatergic), GABA (GABAergic), ChAT (Cholinergic), and 5-HT (Serotonergic) in NC and NC+ZPAK iNs.

DOI: https://doi.org/10.7554/eLife.41356.013

**Figure supplement 3.** Gene set expression analysis.

DOI: https://doi.org/10.7554/eLife.41356.014

**Figure supplement 4.** Immunocytochemical analysis of MYH3 expression in L1, L4 and L8 NC and NC+ZPAK iNs.

DOI: https://doi.org/10.7554/eLife.41356.015

**Figure supplement 5.** Fold change of genes activated or suppressed by ZPAK assessed by SYBR qPCR in L2, L5, and L7 at D5, 10, 15, and 20 of conversion relative to GAPDH.

DOI: https://doi.org/10.7554/eLife.41356.016

**Figure supplement 6.** Calcium imaging reveals increased spontaneous activity in NC+ZPAK iNs.

DOI: https://doi.org/10.7554/eLife.41356.017

---

*NCAM1/2* (Neural Cell Adhesion), *MNX1* (Neural Homeobox), *MAP2* (Neural Microtubule Protein), and *CAMK2* (Central Nervous System Kinase) in NC+ZPAK but lower expression levels of cyclins such as *CCND2* (Cyclin D2) as well as the myogenic factors *MEF2* (Myocyte Enhancer Factor) and *MYOG* (Myogenin), which are known to limit fibroblast to neuronal reprogramming, in NC+ZPAK (*Figure 3F*, *Figure 3—figure supplement 3*, *Figure 3—figure supplement 4*).

Next, gene ontology (GO) enrichment analysis revealed that genes upregulated in NC (log2FC > 1) were only enriched for three neuronal GO terms - GO:0034220 Ion transmembrane transport, GO:0006811 ion transport, and GO:0099537 Trans-synaptic signaling - whereas GO terms enriched in genes upregulated in NC+ZPAK included many GO terms categorized for neural development (GO:0021953 Central Nervous System Neuron Differentiation, GO:0007272/0008366 Ensheathement of Neurons/Axons, GO:0021700 Developmental Maturation), synapse development (GO:0007416 Synapse Assembly, GO:0050808 Synapse Organization, GO:0007268 Chemical Synaptic Transmission), neural activity (GO:0001964 Startle Response), and membrane potential maintenance (GO:0006811,0098660,0006811 Regulation of Ion Transport, Inorganic Ion Transport, Ion Transport) (*Figure 3G*, *Figure 3—figure supplement 1*; *Supek et al., 2011*). When comparing the average and median expression of the genes in these GO terms, we observed consistent upregulation in NC+ZPAK (*Figure 3—figure supplement 1*). These data indicate that NC+ZPAK yields iNs with a more defined and probably more human brain-like transcriptional profile. To assess neuronal enrichment in a threshold-free manner, and to quantify overlap of expression with adult brain, we employed the rank-rank hypergeometric overlap (RRHO) test (*Plaisier et al., 2010*). Briefly, this method generates a map of the transcriptional overlap between any two systems by comparing two ranked lists of differentially expressed genes. We applied this method to compare the overlap of NC

versus NC-ZPAK to fibroblast versus adult brain (Allen Institute) to evaluate the extent to which either condition more closely resembled adult brain gene expression. Using RRHO, we observed a pronounced bias in the overlap of NC-ZPAK with adult brain, indicating that, relative to NC, NC-ZPAK iNs were indeed more similar to brain expression patterns (*Figure 3H*). To make a direct comparison of the functional maturity of NC to NC+ZPAK iNs, we performed calcium imaging of NC and NC+ZPAK iNs, revealing that NC+ZPAK iNs have more spontaneous calcium transients than iNs cultured in NC alone (*Figure 3—figure supplement 5*). As calcium transients have been established as a reliable readout of neural activity in vitro, this direct comparison between NC and NC+ZPAK iNs provides further evidence that ZPAK is producing a more mature and defined neuronal state (*Rosenberg and Spitzer, 2011*). Taken together, these data indicate that NC-ZPAK iNs possess a transcriptional profile more closely resembling that of mature neurons and that the original fibroblast transcriptional program, as well as other non-neuronal directions, were substantially reduced or absent.

To gain a better understanding of how each of the four pathway modulators improves iN conversion, we sought to obtain a more detailed understanding of the individual processes influenced by ZM, PY, AZ, and KC.

## Inhibition of JAK2 removes fibroblasts from the cell cycle and promotes mesenchymal-to-epithelial transition (MET)

Removal of fibroblasts from the cell cycle has been reported to improve neuronal reprogramming (*Jiang et al., 2015*; *Liu et al., 2013*). Based on the enrichment of the IGF-1 signaling and the specific transcriptional footprint called by the IPA software, we selected the JAK2 inhibitor AZ as a promising and well-characterized inhibitor, because IGF1/IGF1R is known to enhance cell cycle entry of fibroblasts (*Mairet-Coello et al., 2009*; *Gross and Rotwein, 2016*) and signals through the JAK2 substrate STAT3 (*Xiong et al., 2014*; *Nosaka et al., 1999*; *Zhang and Derynck, 1999*) (*Figure 4A*). We therefore asked whether AZ removed fibroblasts from the cell cycle to enable direct conversion. As expected, Western blot analysis for STAT3 revealed that converting fibroblasts cultured in NC showed a decrease in STAT3 within the first six days of conversion, and the addition of AZ strongly promoted this decrease (*Figure 4B*). Using carboxyfluorescein succinimidyl ester (CFSE) to examine cell divisions by flow cytometry, we consistently detected that fibroblasts cultured with 0.1 µM AZ or ZPAK went through significantly fewer divisions than fibroblasts cultured in control medium (*Figure 4C,D*). Importantly, flow cytometry for DAPI staining did not detect elevated toxicity in AZ-treated cells. Change in the cell phenotype from multipolar mesenchymal cells to polarized epithelial cells is an important developmental process known as the mesenchymal-epithelial transition (MET) (*Chaffer et al., 2007*). MET machinery plays an important role during direct conversion of fibroblasts to neurons (*He et al., 2017*), and p-STAT3 dimers directly promote epithelial-to-mesenchymal transition (EMT)-related gene expression (*Wendt et al., 2014*). Perturbations in STAT3 signaling have been linked to MET (*Zhang et al., 2014*; *Renjini et al., 2014*). Therefore, we monitored the E-Cadherin to N-Cadherin switch during iN conversion in NC and in NC + AZ (*Nakajima et al., 2004*; *Lamouille et al., 2014*) and found that AZ induced a more rapid switch towards E-Cadherin (*Figure 4D*), a switch that is a classic hallmark of MET (*Scarpa et al., 2015*; *Zhang et al., 2013*).

## Activation of integrin and rho signaling promotes neuronal complexity

The morphological changes a fibroblast has to undergo to transform into a neuron are substantial, and we have observed a pronounced increase in neuronal complexity in ZPAK compared to NC (*Figure 2B*). To determine which ZPAK molecules primarily facilitate these structural modifications, we used ßIII-tubulin-based tracing and scoring to assess the neuronal complexities of iN cultures derived in NC containing either PY, ZM, AZ, or KC; we found that both PY and ZM produced significantly increased complexity scores (*Figure 4E–F*). Interestingly, PY was shown previously to promote integrin β1 stability and signaling of human pluripotent stem cells (*Xu et al., 2010*), and we reasoned that PY also enhanced integrin-dependent attachment to extracellular matrices of iNs, thereby promoting neurite outgrowth and interactions (*Figure 4G*). Based on the fact that ZM is a Raf-1 activator, we reasoned that the structural complexity increase in NC + ZM was associated with increased F-actin activity (*Figure 4G*). Thus, we stained fibroblasts cultured with ZM or PY for 48 hr with fluorescent phalloidin, which revealed a marked increase in phalloidin signal by ZM and also a significant

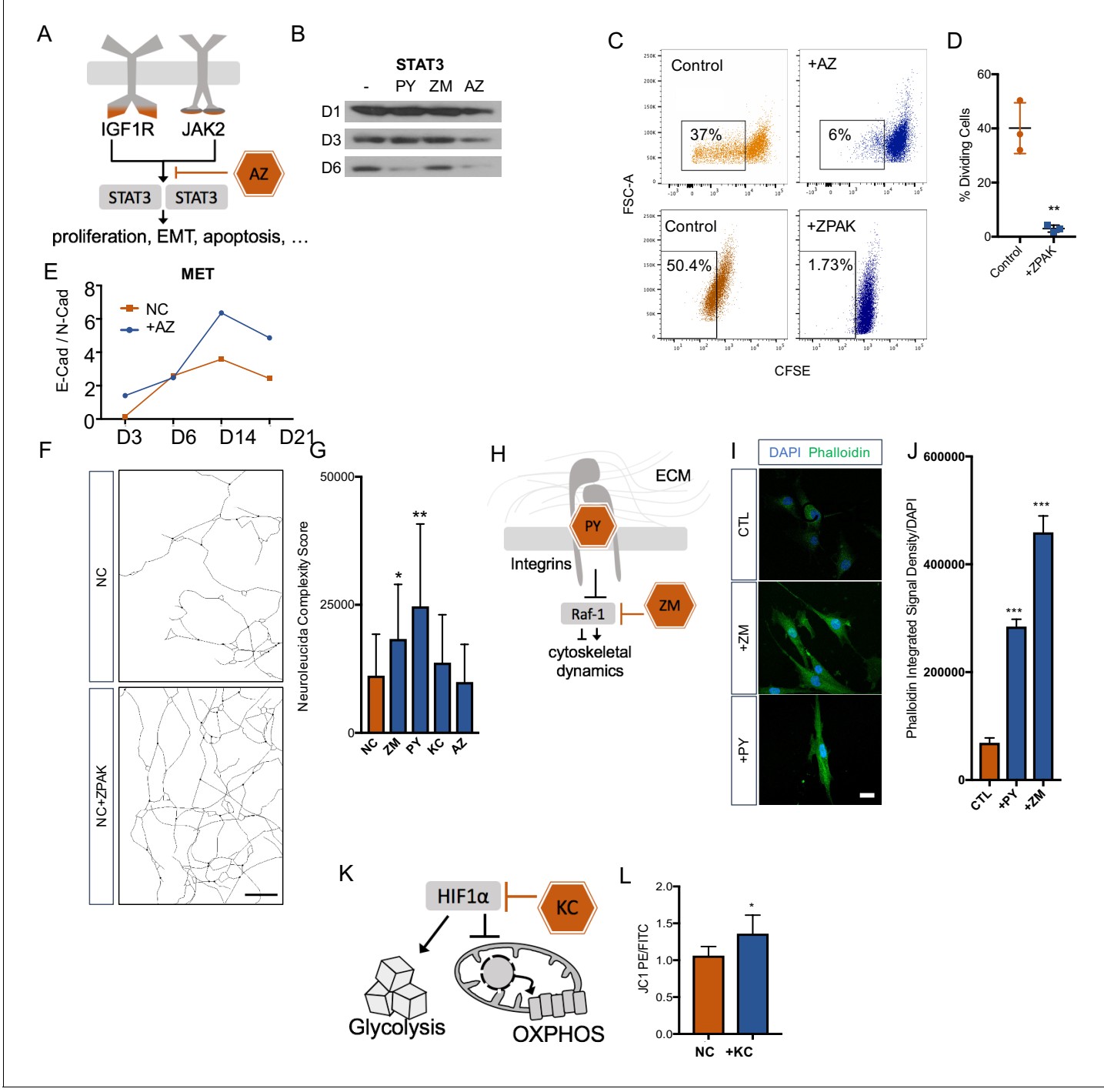

**Figure 4.** Cellular mechanisms influenced by ZPAK-mediated reprogramming. (**A**) Schematic diagram of AZ targeting confluence of IGF1R and JAK2 signaling pathways. (**B**) Western blot analysis of protein levels of STAT3 assessed at one, three, and six days in L6 fibroblast media containing PY, ZM, AZ or control media. (**C**) CFSE stained L1, L5, and L8 fibroblasts cultured with control, AZ-containing medium, or ZPAK-containing medium for 48 hours after plating at 50% confluency. Boxes indicate percentages of cells that have undergone at least one cellular division. (**D**) Quantification of % dividing cell reduction in ZPAK from C. Results are shown as mean ± SD. n = 3, ** P < 0.01. Significance values were calculated by t-test. (**E**) Two color flow cytometry analysis of E- cadherin (E-Cad) and N-cadherin (N-Cad) expression in L10 and L6 fibroblast-to-iN conversion in NC or NC+AZ. Increasing ratios of E-Cad/N-Cad are indicative of the mesenchymal-to-epithelial switch. (**F**) Representative Neurolucida reconstruction of L1 reprogrammed for 21 days in NC or NC+ZPAK medium. Scale bar 100 μm (**G**) Neurolucida complexity scores of iNs derived from NC or NC+ZPAK components. Complexity scores were normalized to cell number by counterstaining with DAPI to count cell bodies. Results are shown as mean ± SD. n = 3, * P < 0.05. (**H**) Schematic diagram of PY and ZM interaction with cytoskeletal dynamics. (**I**) L1 fibroblasts cultured for 48h with ZM, PY, or CTL medium labeled for

*Figure 4 continued on next page*

*Figure 4 continued*

F-Actin with FITC Phallodidin and nuclei with DAPI. Scale bar 20 µm. (J) Integrated signal density of FITC phalloidin stain from H. Signal density was normalized to cell numbers by DAPI. Results are shown as mean ± SEM. n = 3, *** P < 0.001. (K) Schematic diagram of KC targeting HIF1α to inhibit glycolysis and promote oxidative phosphorylation (OXPHOS). (L) Mitochondria in L10, L4, & L2 iNs cultured for 21 days in NC or NC+KC stained with JC-1 and measured for membrane depolarization by flow cytometry. Increased ratios of aggregate (PE) to diffuse (FITC) JC-1 are indicative of increased mitochondrial membrane potential. Results are shown as mean ± SD. n = 3. Significance values were calculated by t-test.
DOI: https://doi.org/10.7554/eLife.41356.019
The following figure supplement is available for figure 4:

**Figure supplement 1.** Mitochondria in L10, L4, and L2 iNs cultured for one or two weeks in NC or NC + KC, stained with JC-1 and measured for membrane depolarization by flow cytometry.
DOI: https://doi.org/10.7554/eLife.41356.020

increase by PY (*Figure 4H–I*). These results are consistent with the reported roles of Raf-1 activation and actin polymerizations in organizing cytoskeletal shape and neuronal morphology (*Ehrenreiter et al., 2005*; *Hsueh, 2012*; *Bosch et al., 2014*), and they suggest that ZM and PY promote morphological rearrangements that occur during direct neuronal reprogramming by different but overlapping means.

## Inhibition of HIF1α signaling promotes oxidative phosphorylation in iNs

Consistent with our timeline transcriptome data, the transcription factor HIF1α has been reported to be one of the top downregulated factors during direct neuronal conversion from a variety of originating cell types (*Omrani et al., 2018*). Increased oxidative phosphorylation is a hallmark of neuronal identity, and glycolysis has been reported to limit neural reprogramming in many protocols (*Zheng et al., 2016*; *Gascón et al., 2016*; *Agostini et al., 2016*). As HIF1α is a major inhibitor of oxidative phosphorylation (OXPHOS) and a mediator of glycolysis, we hypothesized that KC, a HIF1α inhibitor, improved iN conversion by facilitating OXPHOS (*Figure 4J*). Using the JC1 dye, a cationic dye that accumulates in energized mitochondria with high membrane potentials (*Smiley et al., 1991*), we found that cultures at one, two and three weeks of conversion had higher mitochondrial membrane potentials in the presence of KC at all time points and significantly higher mitochondrial membrane potentials by week three (*Figure 4K*, *Figure 4—figure supplement 1*). These data are consistent with a HIF1α blockade initiated by KC promoting the metabolic switch towards OXPHOS that is necessary for iN conversion.

## Epigenetic signatures of donor age are preserved in ZPAK-derived iNs

One unique characteristic of direct iN conversion compared to iPSC reprogramming and subsequent neuronal differentiation is the retention of the cellular marks of aging (*Mertens et al., 2015*; *Tang et al., 2017*; *Victor et al., 2018*; *Liu et al., 2016*). As our additional ZPAK pathway modulators might impact the cellular age of the derived iNs, we sought to verify that ZPAK did not affect the epigenetic age of derived iNs based on age-dependent DNA methylation (*Jones et al., 2015*). Similar to other global aging features, age-dependent DNA methylation of CpGs has been shown to be preserved in iNs (*Huh et al., 2016*) and reverts to a predicted prenatal age even in iPSCs from donors older then 90 years of age (*Lo Sardo et al., 2017*). We quantified CpG methylation for 850,000 sites for two young (0 and 1 years), two middle age (both 29 years), and two old (71 and 87 years) purified iN cultures in NC+ZPAK, as well as one young (1 year) and one old (87 years) unconverted fibroblast culture. Based on the top 5000 differentially methylated regions identified using ChAMP (*Morris et al., 2014*), we observed a clear difference between young and old donors, with middle-age donors showing an intermediate age-dependent CpG methylation pattern (*Figure 5A*). Most importantly, the methylation patterns of the young and old NC+ZPAK iNs were largely unchanged compared to their parental fibroblasts (*Figure 5A*). To apportion the majority of the variation, principal component analysis (PCA) based on all 850,000 CpGs consistently clearly separated the samples with respect to their age as the strongest component (*Figure 5B*). Taken together, these results indicate that age-dependent DNA methylation patterns are maintained during ZPAK-assisted iN conversion and that NC + PAKZ is suitable to efficiently generate human neuronal models that recapitulate age-associated epigenetic changes.

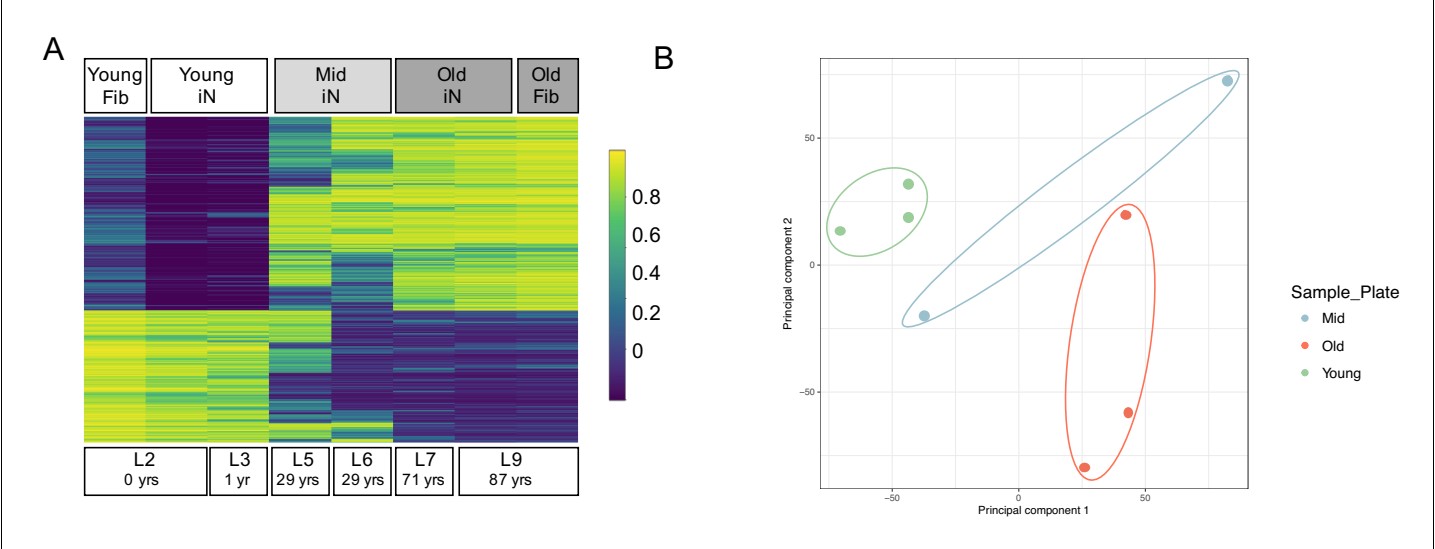

**Figure 5.** Epigenetic signatures of donor age are preserved in ZPAK-derived iNs. (**A**) Heat map showing the top 5000 significantly (padj < 0.05) differentially methylated CpGs between iNs generated by NC+ZPAK from two young (0, 1 years), two middle age (29 years), and two old (71, 87 years) donors, and one young and one old untreated fibroblast (0, 87 years, respectively). Each line represents a single probe. High methylation levels are shown in yellow, low methylation is shown in blue. Methylation of one young and one old paired, unconverted fibroblast is shown next to its iN counterpart. (**B**) Principal Component Analysis (PCA) of the methylation profiles of two young iNs, one young fibroblast, two middle age iNs, two old iNs, and one old fibroblast cell line. All iNs were reprogrammed with NC+ZPAK medium. Plot shows principal component one and principal component two for each sample. Samples closer to each other in principal component space are similar in their methylation profiles.
DOI: https://doi.org/10.7554/eLife.41356.021

The following figure supplement is available for figure 5:

**Figure supplement 1.** High passage iN conversion in NC and NC+ZPAK.
DOI: https://doi.org/10.7554/eLife.41356.022

## Discussion

Significant gaps remain in our understanding of the mechanisms influencing neuronal specification during direct reprogramming. Consequently, iN yields from current protocols are often paltry, necessitating large numbers of converting cells to get sufficient successfully reprogrammed iNs for experimental analysis. Further, improvements to these protocols are largely based on trial and error, with few predictive tools available. In this study, we leveraged RNAseq data to uncover several key molecular events that can be targeted with small molecules to improve iN yield. These modifications, combined with our streamlined lentiviral system, provide a state-of-the-art improvement in current iN direct reprogramming protocols and shed insight into mechanisms driving neuronal specification during reprogramming.

Efficiently overcoming competing cellular programs has long been a challenge in the somatic reprogramming field. Here, we report that ZPAK causes a more thorough elimination of competing fibroblast programs on the transcriptional level than classic medium, as represented by both upregulation of neuron-specific genes and, importantly, downregulation of myogenic and fibroblast-specific genes known to limit neuronal specification during late reprogramming. These transcriptional changes are accompanied by other important phenotypes associated with neuronal fate, including removal of cells from the cell cycle, increased mitochondrial membrane potential, increased GCAMP activity, and enhanced mesenchymal to epithelial plasticity. There has also been an increased recognition of the important role of cytoskeleton remodeling during somatic cell reprogramming (*Sakurai et al., 2014*; *Guo et al., 2014*). Neurons have elaborate cytoskeleton structures that are highly specialized and critical for proper function. As the morphological changes from a fibroblast to a neuron are substantial, we were interested to discover that two of our compound modifications, ZM and PY, were associated with increased neuronal morphological complexity in addition to increasing iN yield. This phenotype was accompanied by an increase in F-Actin activity, a potent component of cellular plasticity. Cytoskeleton remodeling is associated with sheer stress, and the

actin cytoskeleton can sense and respond to these stresses with apoptotic signals. Therefore, we propose that the inclusion of ZM and PY in direct neuronal reprogramming could promote cytoskeleton reorganization and spare converting fibroblasts from mechanical stress-induced apoptosis. Our findings illustrate the important contribution actin structure makes during reprogramming and suggests that increased cellular structural plasticity could be a fruitful strategy for improving neuronal lineage commitment during reprogramming.

Classic direct conversion protocols, which we and others have used to differentiate fibroblasts to mature human neurons, are highly inefficient and fail to successfully reprogram the majority of input cells. Additionally, a significant heterogeneity exists in the reprogrammed pool, with many cells existing in a state that is not quite fibroblast, not quite neuron. Although many relevant phenotypes have been found between patient and control neuronal cultures reprogrammed in currently used media, we predict that new phenotypes might be revealed from studying neurons in conditions that more thoroughly commit fibroblasts to neuronal fate specification and, in turn, might lead to the discovery of more effective treatments for neurological disorders. Further, as iNs have shown promise in cell replacement therapies, we propose that using culture conditions that completely commit fibroblasts to the neuronal lineage will result in an increased therapeutic potential of iNs for possible neuronal replacement. Importantly, ZPAK improves iN yields even at very late fibroblast passages that could be required to produce therapeutic numbers of cells (*Figure 5—figure supplement 1*). Although 100% efficiency in establishing neuronal specification remains elusive, the development of new neuronal reprogramming techniques, such as UNA and ZPAK media, takes us one step closer to this goal.

## Materials and methods

**Key resources table**

| Reagent type (species) or resource | Designation | Source or reference | Identifiers | Additional information |
|---|---|---|---|---|
| Chemical compound, drug | AZ960 | Cayman | RRID:SCR_008945 | |
| Chemical compound, drug | Pyrintegrin | Tocris | RRID:SCR_003689 | |
| Chemical compound, drug | ZM336372 | Cayman | RRID:SCR_008945 | |
| Chemical compound, drug | KC7F2 | Fischer Scientific | RRID:SCR_008452 | |
| Chemical compound, drug | Phalloidin CruzFluor 488 Conjugate | SCBT | RRID:AB_2631056 | '1:1000' |
| Antibody | N-Cadherin (Mouse monoclonal) | Miltenyi Biotec | RRID:AB_2727381 | '1:50' |
| Antibody | E-Cadherin (Mouse monoclonal) | Miltenyi Biotec | RRID:AB_10827695 | '1:11' |
| Antibody | STAT3 (Rabbit monoclonal) | Cell Signaling | RRID:AB_2716836 | '1:1000' |
| Antibody | Anti-NeuN (Mouse monoclonal) | EMD Millipore | RRID:AB_2298772 | '1:200' |
| Antibody | Anti-Tubulin β−3 (Rabbit polyclonal) | Covance | RRID:AB_2313773 | '1:3000' |
| Antibody | Anti-GABA (Rabbit polyclonal) | Sigma | RRID:AB_477652 | '1:500' |
| Antibody | Anti-vGlut1 (Rabbit polyclonal) | Synaptic Systems | RRID:AB_887875 | '1:100' |
| Antibody | Anti-TH (Rabbit polyclonal) | Pel-Freez | RRID:AB_2313713 | '1:250' |

*Continued on next page*

*Continued*

| Reagent type (species) or resource | Designation | Source or reference | Identifiers | Additional information |
|---|---|---|---|---|
| Antibody | Anti-ChAT (Goat polyclonal) | EMD Millipore | RRID:AB_10615776 | '1:100' |
| Antibody | Anti-5-HT (Rabbit polyclonal) | Protos Bio Tech | RRID:AB_2313881 | '1:1000' |
| Antibody | Anti-PSANCAM (Mouse monoclonal) | Miltenyi Biotec | RRID:AB_1036069 | '1:50' |
| Antibody | anti-MYH (Mouse monoclonal) | Santa Cruz Biotechnology | RRID:AB_10989398 | '1:100' |
| Chemical compound, drug | Cell Trace CFSE | ThermoFisher Scientific | C34554 | |
| Commercial assay or kit | Mito Probe JC-1 | ThermoFisher Scientific | M34152 | '1:1000' |
| Software, algorithm | Ingenuity Pathway Analysis | QIAGEN Bioinformatics | RRID:SCR_008653 | |

## Direct conversion of human fibroblasts into iNs

Primary human dermal fibroblasts from donors between 0 and 88 years of age were obtained from the Coriell Institute Cell Repository, the University Hospital in Erlangen and Shiley-Marcos Alzhiemer's Disease Research Center (*Figure 2—source data 2*). Protocols were previously approved by the Salk Institute Institutional Review Board and informed consent was obtained from all subjects. Fibroblasts were cultured in DMEM containing 15% tetracycline-free fetal bovine serum and 0.1% NEAA (Thermo Fisher Scientific), transduced with lentiviral particles for EtO and XTP-Ngn2:2A:Ascl1 (E + N2A), or the combined tetOn system cassette consisting of the rtTAAdv. [Clonech] driven by the UbC promoter, Ngn2:2A:Ascl1 under control of the TREtight promoter [Clontech], and a puromycin-resistance gene driven by the PGK promoter (UNA, *Figure 1A*) and expanded in the presence of G418 (200 μg/ml; Thermo Fisher Scientific) and puromycin (1 μg/ml; Sigma Aldrich), or puromycin only, respectively, as 'iN-ready' fibroblast cell lines. Following at least three passages after viral transduction, 'iN-ready' fibroblasts were trypsinized and pooled into high densities (30.000–50.000 cells per $cm^2$; appx. a 2:1 – 3:1 split from a confluent culture) and, after 24 hr, the medium was changed to neuron conversion (NC) or NC+ZPAK medium based on DMEM:F12/Neurobasal (1:1) for three weeks. NC contains the following supplements: N2 supplement, B27 supplement (both 1x; Thermo Fisher Scientific), doxycycline (2 μg/ml, Sigma Aldrich), Laminin (1 μg/ml, Thermo Fisher Scientific), dibutyryl cyclic-AMP (500 μg/ml, Sigma Aldrich), human recombinant Noggin (150 ng/ml; Preprotech), LDN-193189 (5 μM; Fisher Scientific Co) and A83-1 (5 μM; Santa Cruz Biotechnology Inc), CHIR99021 (3 μM, LC Laboratories), Forskolin (5 μM, LC Laboratories) and SB-431542 (10 μM; Cayman Chemicals). ZPAK contains the following supplements: Pyrintegrin (1 μM; Tocris), ZM336372 (0.175 μM; Cayman), AZ960 (0.1 μM; Cayman), and KC7F2 (7.5 μM; Fischer Scientific). Medium was changed every third day. For further maturation up to six weeks, iNs were switched to BrainPhys (STEMCELL Technologies)-based neural maturation media (NM) containing N2, B27, GDNF, BDNF (both 20 ng/ml, R and D), dibutyryl cyclicAMP (500 μg/ml, Sigma Aldrich), doxycycline (2 μg/ml, Sigma-Aldrich) and laminin (1 μg/ml, Thermo Fisher Scientific). For maturation on astrocytes for morphological analysis and calcium imaging, iNs were carefully trypsinized during week four and replated on a feeder layer of mouse astrocytes and cultured in NM media containing 1% KOSR (Thermo Fisher Scientific).

## Whole transcriptome mRNA sequencing and methylation array

Total bulk RNA was extracted from fibroblasts and converting iNs at all collected time points using Trizol LS reagent (Thermo Fischer). RNA integrity (RIN) numbers were assessed using the Agilent TapeStation before library preparation. RNA-Seq libraries were prepared using the TruSeq Stranded mRNA Sample Prep Kit according to the manufacturer's instructions (Illumina). Libraries were sequenced single-end 50 bp using the Illumina HiSeq 2500 platform. Read trimming was performed using TrimGalore, read mapping was performed using STAR, raw counts were generated using

HTseq variance stabilizing transformation normalization (vst) and differential expression analysis was performed in DEseq2. Pathway and network analysis was performed using Ingenuity Pathway Analysis (Qiagen) from FPKM normalized HTseq generated gene counts under the time course analysis module. Pathways with a Z-Score $\geq 3$ (99% confidence interval) were considered for further study.

Total genomic DNA was extracted from bulk fibroblasts and flow cytometry sorted PSA-NCAM + 21 day iNs as described below with the DNeasy Blood and Tissue Kit (Qiagen). DNA methylation assays were performed on the MethylationEPIC BeadChip as per the standard manufacturer's protocol (Illumina). Raw intensity idat files were processed and analyzed using the R packages ChAMP (*Morris et al., 2014*) or Rnbeads (*Assenov et al., 2014*); arrays were normalized using the BMIQ procedure (*Teschendorff et al., 2013*).

### Flow cytometry assessment of PSA-NCAM, ECAD/NCAD, Cell Proliferation, and Mitochondrial Membrane Polarization

For PSA-NCAM analysis, iNs were detached using TrypLE and stained for PSA-NCAM directly conjugated to PE (Miltenyi Biotec; 1:50) for 1 hr at 4°C in sorting buffer (250 mM myo-inositol and 5 mg/ml polyvinyl alcohol, PVA, in PBS) containing 1% KOSR. Cells were washed and resuspended in sorting buffer containing EDTA and DNAse and filtered using a 40 µm cell strainer. For co-staining with NCAD and ECAD, cells were detached and prepared as above and stained for N-CAD:APC (Miltenyl Biotec, CD325, 1:20) and E-CAD:PE (Miltenyl Biotec, CD324, 1:11). To determine the effect of AZ960 on proliferation of fibroblast cell lines, the cellular divisions were quantified with the CellTrace CFSE proliferation assay (Thermo Fisher). $1.2 \times 10^6$ fibroblasts at full density on a single well of a six-well plate (9 cm$^2$) were split into two 60 mm plates (21 cm$^2$) and proliferated in DMEM containing 15% tetracycline-free fetal bovine serum and 0.1% NEAA (Life Technologies), 2.5 µM of CellTrace CFSE proliferation dye (Invitrogen), and 0.1 µM AZ960 for 48 hr. Mitochondrial membrane potential was analyzed using the MitoProbe JC-1 assay kit (Thermo Fisher, M34152). In all cases analysis was conducted on the FACS Canto II platform.

### Image collection and analysis

Cells were transferred to tissue culture-treated ibidi µ-slides for imaging. Cells were fixed with 4% PFA for 20 min at room temperature and washed 3 $\times$ 15 min with TBS, followed by a 1 hr block with TBS + 4% bovine serum albumin and 0.1% Triton X-100 (TBS++). Primary antibodies (Anti-NeuN, 1:200, EMD Millipore; Anti-ßIII-tubulin, 1:3000, Covance; Phalloidin CruzFluor 488 Conjugate, 1:1000, SCBT) were applied overnight at 4°C. After washing as described above, samples were incubated in 1:250 donkey anti-mouse, chicken, or rabbit secondary antibody for 2 hr at room temperature. Nuclear staining was done with DAPI (1/10,000; Sigma-Aldrich). After washing, slides were mounted in PVA-DAPCO (Sigma Aldrich). Confocal images were taken on Zeiss LSM780 or Zeiss AiryScan series confocal microscope. All data for one experiment were acquired from cells cultured and processed in parallel on the same microscope with the exact same setting reused. For analysis, 2 µm confocal sections through the nuclear layer were acquired from three confocal z stacks. Neurolucida software was used for manual tracing of entire neuronal processes, and data were analyzed using NeuroExplorer (MicroBrightField Inc). All tracings were performed in a blinded manner. For phalloidin staining, automatic thresholding in ImageJ was used to binarize the images and green fluorescence intensity was calculated minus background intensity. At least 50 different cells were analyzed in each experiment, and the mean $\pm$ SEM optical density (OD) was then calculated.

### Calcium Imaging

iNs converted in NC or NC+ZPAK were transduced with lentiviral particles for CAG-GCAMP5G and LV-hSyn-dsRed at 21 days of conversion and replated on astrocytes for maturation as described above. Calcium imaging were performed four weeks after plating on astrocytes. Imaging was performed in BrainPhys media (Stemcell Technologies) on a Yokogawa Cell Voyager 1000 Spinning Disk Confocal Microscope. We only analyzed 10 cells per field that were hSyn-dsRed + and exhibited neuron identity. Calcium responses were calculated as the change in fluorescence intensity ($\Delta F$) over the initial fluorescence intensity $(F-F_0)/F_0$, in which F is the fluorescence at a given time point and $F_0$ was calculated as the average of the first five inactive fluorescence measurements at the start of imaging. A non-response area for each recording was measured for background subtraction, and

imaging bleach was corrected for using the ImageJ plugin Fiji (NIH, Bethesda, MD). The threshold for a positive calcium event was defined as local maxima when fluorescence response within a soma exceeded a value greater than three standard deviations above the mean.

## SYBR qPCR

Bulk mRNA was extracted as described above, and 1 µg of RNA was then reverse transcribed using the Superscript III Reverse Transcriptase kit (Thermo Fisher). Quantitative differences in gene expression were determined by real-time qPCR using SYBR Green Master Mix (Bio-Rad) and a spectrofluorometric thermal cycler (CFX384, Bio-Rad). Values are presented as the ratio of target mRNA to GAPDH expression obtained for each time point and treatment. Primer sequences used are the following:

    MNX1: GATGCCCGACTTCAACTCCC, GCCGCGACAGGTACTTGTT
    MAP2: CTCAGCACCGCTAACAGAGG, CATTGGCGCTTCGGACAAG
    CCND2:TTTGCCATGTACCCACCGTC, AGGGCATCACAAGTGAGCG
    MYOG: GGGGAAAACTACCTGCCTGTC, AGGCGCTCGATGTACTGGAT
    MEF2C: CTGGTGTAACACATCGACCTC, GATTGCCATACCCGTTCCCT
    NCAM1: GTCCTGCTCCTGGTGGTTGT, TGACCGCAATGCACATGAA
    NCAM2: GACGTGCCATCCAGTCCCTA, ATGGGAGTCCGGTTTGTTGA
    CAMK2A: AACCTTGGCTCCAGCATGAA, AAGGGAGACAGGAGGCCTTG
    GAPDH: TGCACCACCAACTGCTTAGC, GGCATGGACTGTGGTCATGAG

## Western blot analysis

Cell lysates were prepared in Lysis buffer A (20 mM Tris pH 7.5, 100 mM NaCl, 1 mM EDTA, 2 mM EGTA, 50 mM β -glyceropshate, 50 mM NaF, 1 mM sodium vanadate, 2 mM dithiothreitol, proteinase inhibitor cocktail (Roche) and 1% Triton X-100 and subjected to Western blot according to the standard procedures. The primary antibody used was rabbit mAb STAT3 (1:1000, Cell Signaling, 79D7, #4904).

## Statistical analysis

Statistical values for RNA-Seq and CpG methylation data were corrected for false discovery rates (FDR) using the Benjamini-Hochberg method implemented in R. Statistical tests of quantitative data were calculated using GraphPad Prism seven software, with the method indicated for each figure. Significance evaluation are marked as $*p<0.05$; $**p<0.01$; $***p<0.005$ and $****p<0.001$.

## Acknowledgements

We thank the Coriell Institute, Johannes Schlachetzki and Jurgen Winkler at the University Hospital in Erlangen, and the Shiley-Marcos Alzheimer's Disease Research Center at the University of California, San Diego (UCSD) for primary human fibroblasts, and Mary Lynn Gage for editorial comments. The study was supported by the Paul G. Allen Family Foundation, the National Institute on Aging (R01-AG056306-01 and K99-AG056679-01), the National Cancer Institute (P30 CA014195), the Austrian Science Fund FWF (SPIN doctoral school), the JPB Foundation, the Glenn Foundation Center for Aging Research, the American Federation for Aging Research (AFAR), the Leona M and Harry B Helmsley Charitable Trust (2012- PG-MED002), Annette Merle-Smith, CIRM (TR2-01778), The G Harold and Leila Y Mathers Charitable Foundation, and the 2014 NARSAD Young Investigator Grant from the Brain and Behavior Research Foundation.

## Additional information

### Competing interests

Fred H Gage: Reviewing editor, *eLife*. The other authors declare that no competing interests exist.

## Funding

| Funder | Grant reference number | Author |
| --- | --- | --- |
| Austrian Science Fund | | Jerome Mertens |
| Paul G. Allen Family Foundation | | Fred H Gage |
| National Institute on Aging | R01-AG056306-01 | Fred H Gage |
| JPB Foundation | | Fred H Gage |
| Glenn Family Foundation | | Fred H Gage |
| American Federation for Aging Research | | Fred H Gage |
| Leona M. and Harry B. Helmsley Charitable Trust | 2012- PG-MED002 | Fred H Gage |
| Annette Merle-Smith CIRM | TR2-01778 | Fred H Gage |
| G Harold and Leila Y. Mathers Foundation | | Fred H Gage |
| Brain and Behavior Research Foundation | 2014 NARSAD Young Investigator Grant | Fred H Gage |
| National Institute on Aging | K99-AG056679-01 | Fred H Gage |
| National Cancer Institute | P30 CA014195 | Fred H Gage |

The funders had no role in study design, data collection and interpretation, or the decision to submit the work for publication.

## Author contributions

Joseph Herdy, Conceptualization, Data curation, Formal analysis, Validation, Investigation, Visualization, Methodology, Writing—original draft, Project administration, Writing—review and editing; Simon Schafer, Conceptualization, Methodology, Writing—review and editing; Yongsung Kim, Conceptualization, Formal analysis, Visualization, Writing—review and editing; Zoya Ansari, Dina Zangwill, Formal analysis, Investigation, Visualization; Manching Ku, Formal analysis, Investigation, Visualization, Methodology; Apua Paquola, Data curation, Software, Investigation, Visualization; Hyungjun Lee, Formal analysis, Validation; Jerome Mertens, Conceptualization, Supervision, Funding acquisition, Visualization, Methodology, Writing—original draft, Project administration, Writing—review and editing; Fred H Gage, Conceptualization, Resources, Supervision, Funding acquisition, Project administration, Writing—review and editing

## Author ORCIDs

Joseph Herdy (iD) http://orcid.org/0000-0002-8636-8134
Manching Ku (iD) http://orcid.org/0000-0001-5168-4308
Fred H Gage (iD) http://orcid.org/0000-0002-0938-4106

## Decision letter and Author response

Decision letter https://doi.org/10.7554/eLife.41356.035
Author response https://doi.org/10.7554/eLife.41356.036

# Additional files

## Supplementary files

• Transparent reporting form DOI: https://doi.org/10.7554/eLife.41356.023

## Data availability

Sequencing data have been deposited in Annotare under accession codes E-MTAB-7250, E-MTAB-7226, and E-MTAB-7259.

The following datasets were generated:

| Author(s) | Year | Dataset title | Dataset URL | Database and Identifier |
|---|---|---|---|---|
| Herdy JR, Schafer S, Kim Y, Ansari Z, Zangwill D, Ku M, Paquola ACM, Lee H, Mertens J, Gage FH | 2019 | Methylation array of young, mid age, and old induced neurons cultured in NC+ZPAK, as well as unconverted old and young fibroblast. | https://www.ebi.ac.uk/arrayexpress/experiments/E-MTAB-7226 | Annotare, E-MTAB-7226 |
| Herdy JR, Schafer S, Kim Y, Ansari Z, Zangwill D, Ku M, Paquola ACM, Lee H, Mertens J, Gage FH | 2019 | Timeline RNAseq of fibroblast to neuron direction conversion | https://www.ebi.ac.uk/arrayexpress/experiments/E-MTAB-7259 | Annotare, E-MTAB-7259 |
| Herdy JR, Schafer S, Kim Y, Ansari Z, Zangwill D, Ku M, Paquola ACM, Lee H, Mertens J, Gage FH | 2019 | RNA-seq timeline of fibroblast to neuron conversion using traditional NC media and NC+ZPAK media | https://www.ebi.ac.uk/arrayexpress/experiments/E-MTAB-7250 | Annotare, E-MTAB-7250 |

The following previously published datasets were used:

| Author(s) | Year | Dataset title | Dataset URL | Database and Identifier |
|---|---|---|---|---|
| Miller JA, Guillozet-Bongaarts A, Gibbons LE, Postupna N, Renz A, Beller AE, Sunkin SM, Ng L, Rose SE, Smith KA, Szafer A, Barber C, Bertagnolli D, Bickley K, Brouner K, Caldejon S, Chapin M, Chua ML, Coleman NM, Cudaback E, Cuhaciyan C, Dalley RA, Dee N, Desta T, Dolbeare TA, Dotson NI, Fisher M, Gaudreault N, Gee G, Gilbert TL, Goldy J, Griffin F, Habel C, Haradon Z, Hejazinia N, Hellstern LL, Horvath S, Howard K, Howard R, Johal J, Jorstad NL, Josephsen SR, Kuan CL, Lai F, Lee E, Lee F, Lemon T, Li X, Marshall DA, Melchor J, Mukherjee S, Nyhus J, Pendergraft J, Potekhina L, Rha EY, Rice S, Rosen D, Sapru A, Schantz A, Shen E, Sherfield E, Shi S, Sodt AJ, Thatra N, Tieu M, Wilson AM, Montine TJ, Larson EB, Bernard A, Crane PK | 2017 | gene_expression_matrix_2016_03_03 | http://aging.brain-map.org/api/v2/well_known_file_download/502999992 | Aging, Dementia and Traumatic Brain Injury Study, 502999992 |
| Miller JA, Ding SL, Sunkin SM, Smith | 2014 | RNA-Seq Gencode v10 summarized to genes | http://www.brainspan.org/api/v2/well_known_ | BrainSpan Atlas of the Developing Human |

KA, Ng L, Szafer A,
Ebbert A, Riley ZL,
Royall JJ, Aiona K,
Arnold JM, Bennet
C, Bertagnolli D,
Brouner K, Butler S,
Caldejon S, Carey
A, Cuhaciyan C,
Dalley RA, Dee N,
Dolbeare TA, Facer
BA, Feng D, Fliss
TP, Gee G, Goldy
J, Gourley L, Gre-
gor BW, Gu G,
Howard RE, Jochim
JM, Kuan CL, Lau
C, Lee CK, Lee F,
Lemon TA, Lesnar
P, McMurray B,
Mastan N, Mos-
queda N, Naluai-
Cecchini T, Ngo
NK, Nyhus J, Oldre
A, Olson E, Par-
ente J, Parker PD,
Parry SE, Stevens A,
Pletikos M, Reding
M, Roll K, Sandman
D, Sarreal M, Sha-
pouri S, Shapova-
lova NV, Shen EH,
Sjoquist N, Slaugh-
terbeck CR, Smith
M, Sodt AJ, Wil-
liams D, Zöllei L,
Fischl B, Gerstein
MB, Geschwind
DH, Glass IA, Ha-
wrylycz MJ, Hevner
RF, Huang H,
Jones AR, Knowles
JA, Levitt P, Phillips
JW, Sestan N,
Wohnoutka P,
Dang C

file_download/
267666525

Brain, 267666525

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
