## [Decision Letter]

Thank you for submitting your article "Chemical modulation of transcriptionally enriched signaling pathways to optimize conversion of fibroblasts into neurons" for consideration by *eLife*. Your article has been reviewed by three peer reviewers, one of whom is a member of our Board of Reviewing Editors, and the evaluation has been overseen by Marianne Bronner as the Senior Editor. The following individual involved in review of your submission has agreed to reveal her identity: Marisa Karow (Reviewer #2).

The reviewers have discussed the reviews with one another and the Reviewing Editor has drafted this decision to help you prepare a revised submission.

Summary:

The paper by Herdy et al. describes an interesting attempt to improve the process by which neurons are produced directly from patient fibroblasts. The theoretical advantage of such a method, should it be sufficiently efficient, is that it may be capable of preserving some of the epigenetic changes, occurring during aging or other conditions, in the neurons. The authors formulated a method that uses a revised lentiviral vector that drives the expression of two different neurogenic transcription factors along with a combination of small molecule signaling modulators. Overall, the new method appears to result in a substantial improvement in the production of induced neurons, including from a few patient lines that were "low performers" under the original induction conditions.

The three reviewers agree that this manuscript is worth publishing after a set of focused experiments are performed. Most of the comments are directed at achieving a greater understanding of the actions of the small molecules. Some additional information concerning the consistency of the effects of ZPAK on different sets of fibroblasts and on the total scalability of the new differentiation protocol are also requested. Since this is a Resource and Tools paper, we have tried to restrict the requests to ones that we agree will make the protocols more useful and comprehensive.

Essential revisions:

1) Analyzing effects of treatment with single compounds and with ZPAK.

Can the authors clarify:

• Which fibroblast line(s) were used for the analyses in Figure 2 and subsequent figures?

• Whether they tested multiple inhibitors and activators of the same pathways to generate the data for Figure 1. If they tested multiple, did they all work to some degree or were the results more mixed?

• How well ZPAK works across the different patient lines?

• If they compared the effects of ZPAK to those of the individual compounds based on testing single compound concentrations? Were the concentrations chosen from prior literature or from dose-response experiments that they performed? In either case, additional data to ensure that even single compounds at more optimal concentrations wouldn't have comparable effects to those of ZPAK should be included.

2) Understanding the effects of small molecule signaling modulators.

*Reviewer 2:*

Following NC+ZAPK treatment and Ngn2/Ascl1 expression, the fibroblasts adopt a mixed neuronal phenotype as assessed on a molecular level (Figure 3D). Are any of the neuronal branches adopted with higher frequency? Is there a change when switching from NC to NC+ZAPK? Did the authors check for this via immunofluorescence or electrophysiology?

The authors mention the alternative myogenic fate (as induced by Ascl1 alone in murine fibroblasts (Treutlein et al., 2016) being more highly expressed in the NC condition as compared to NC+ZPAK. Could they confirm this via staining?

In Figure 4C the authors show that AZ-treated cells went through significantly fewer divisions. Was cell division assessed in NC+ZAPK versus NC? In Figure 4E it looks like there are more cells in NC+ZAPK.

*Reviewer 3 (reviewer 1 had similar questions to those described clearly below):*

Comparing NC and NC+ZPAK in Figure 3, it appears the authors used bulk fibroblasts undergoing iN conversion at days 5, 10, 15 and 20. However, given the boost in conversion efficiency with ZPAK (Figures 2D-H), could the fact that there are more neurons and fewer fibroblasts in the NC+ZPAK conversion condition compared to the NC condition account for the increased 'adult / defined' neuronal signature? Can the authors rule out a change in cell ratios and ability to detect small transcript changes in a sub-population rather than changes to the neuron population specifically? In order to conclude that NC+ZPAK not only improves efficiency of conversion but also generates neurons that are more 'human brain-like' the authors need to compare NC neurons to NC+ZKAK neurons directly (e.g., through transcriptional analysis of BIII-tub positive neurons, immunophenotyping of neurons to quantify specific markers that are representative of the reported signature, etc.).

In order to support the authors’ conclusions about gene changes over time in Figure 3, key genes from the RNA-seq dataset should be validated by qPCR with error bars. Specifically, showing changes in key neuro, muscle and fibroblast genes over time; key neuronal genes at later time-points that corroborate the more mature phenotype implied from the Allen brain comparison in Figure 3H; glutamatergic and GABAergic genes in order to conclude that they are expressed at roughly similar levels in Figure 3D.

The differences between NC and NC+ZPAK implied in Figures 3G and 3H seem somewhat at odds with the modest transcriptional changes reported, i.e., "NC-ZPAK iNs possess a transcriptional profile more closely resembling that of mature neurons and that the original fibroblast transcriptional program, as well as other non-neuronal directions, were substantially reduced or absent." But, the first paragraph of the subsection “ZPAK induces a defined neuronal transcription that more closely relates to the adult brain transcriptome than NC alone” indicates only 143 genes were significantly differentially expressed between NC and NC+ZPAK pooling all differences across days 5, 10, 15 and 20. How do the authors integrate these two pieces of data?

The authors should also include the number of differentially expressed genes for each time-point and the full list of 143 DEGs, as supplementary figures. Since this is a small/important gene set for this study, it should be readily accessible.

Does the transcript data support the mechanistic data in Figures 4-5? Is there transcriptional evidence to support the signaling pathways induced by ZPAK over the course of conversion? (if so, again, specific gene changes should be validated from the RNA-seq). Upon activation of these signaling pathways, and dramatic differences in neurite complexity shown in Figures 4E-F, again one might expect to see more differentially expressed genes between NC and NC-ZPAK.

3) Usefulness of the new protocol.

The authors are correct to point out that efficiency matters for investigators trying to produce cultures enriched in neurons and for those trying to make large numbers of neurons for proteomic or screening applications. However, by way of comparison, direct differentiation from iPSCs involves generating neurons from pluripotent cells that can be grown in virtually unlimited numbers. In the case of the current method, the numbers of neurons that can be produced from fibroblasts is the product of the number of fibroblasts that can be generated from each patient sample X the efficiency of differentiation. The authors have shown that the efficiency of their method is better, but how well does the method work in fibroblasts that have been passaged sufficiently to give rise to the billions of neurons that may be required? The lines used in this work were already passaged >10 times, which is promising. However, the authors should demonstrate the process will continue to work on these fibroblast lines, especially those derived from older donors, that are passaged sufficiently to produce the requisite large numbers of neurons.

---

## [Author Response]

Essential revisions:1) Analyzing effects of treatment with single compounds and with ZPAK.Can the authors clarify:• Which fibroblast line(s) were used for the analyses in Figure 2 and subsequent figures?

We appreciate the reviewers’ comment and agree that the cell lines used for each figure should be presented more clearly. We have revised our figure legends and added a summary in Figure 1—figure supplement 2, showing which line was for used for which experiment.

• Whether they tested multiple inhibitors and activators of the same pathways to generate the data for Figure 1. If they tested multiple, did they all work to some degree or were the results more mixed?

Based on the pathways identified by the RNA-Seq IPA analysis presented in Figure 2B, we selected and tested a pair of molecules (one inhibitor and one activator) for each pathway. A total of 20 small molecules were tested and details on target, screening concentration, effect on iN, and reference are given in the revised Figure 2—figure supplement 2.

While we did not test multiple small molecules targeting the same pathway component, our strategy revealed that, for the selected four beneficial molecules, the ‘opposite’ compound of the pair always showed a strong negative, or no effect, on iN conversion. We have revised our wording to make this strategy clearer in the revised manuscript (subsection “Time based RNAseq identifies signaling pathways directing iN conversion”).

• How well ZPAK works across the different patient lines?

We agree that the consistency of ZPAK in improving direct conversion across many cell lines is a central and very important point. All presented data regarding ZPAK efficiency in our original manuscript were based on at least three individual fibroblast lines, and we see a consistent increase in conversion efficiency for all lines. To further strengthen this important point, we added new data showing the percentage of PSA-NCAM-positive cells following NK and NK^+^ZPAK conversion of all 10 lines used in the manuscript. The data were added to the new Figure 2—figure supplement 3.

• If they compared the effects of ZPAK to those of the individual compounds based on testing single compound concentrations? Were the concentrations chosen from prior literature or from dose-response experiments that they performed? In either case, additional data to ensure that even single compounds at more optimal concentrations wouldn't have comparable effects to those of ZPAK should be included.

We thank the reviewers for their comment and agree that reporting the optimal concentration of the ZPAK cocktail is important to maximize the conversion benefit of this protocol. Indeed, for our initial screen of the individual compounds, we used concentrations based on literature (typically targeting a 2x EC50 concentration, also see Figure 2—figure supplement 2). However, when the four ZPAK components were combined, we observed slightly adverse effects. We therefore optimized the concentrations for the final ZPAK cocktail accordingly, yielding our ‘final’ ZPAK concentrations (1/4^th^ of full concentrations). For the revised manuscript, we have added new data, presented as Figure 2—figure supplement 4, to make this point clearer. We now show that (A) our final ZPAK cocktail produced more PSA-NCAM-positive iN cells than the ‘full’ or half concentrations; (B) the final ZPAK cocktail is superior to any individual compounds in various concentrations, and (C-D) neuronal morphologies and neuronal marker expression are optimal in the final ZPAK cocktail compared to the full and half concentrations. Further, to increase clarity, we noted the concentrations used in the figure legends when appropriate.

2) Understanding the effects of small molecule signaling modulators.

Reviewer 2:

Following NC+ZAPK treatment and Ngn2/Ascl1 expression, the fibroblasts adopt a mixed neuronal phenotype as assessed on a molecular level (Figure 3D). Are any of the neuronal branches adopted with higher frequency? Is there a change when switching from NC to NC+ZAPK? Did the authors check for this via immunofluorescence or electrophysiology?

We thank the reviewer for their comment and agree that it is important to understand if there is a difference in neuronal subtype fate commitment during conversion induced by ZPAK. Although we saw no significant differences in lineage-specifying mRNAs between NC and NC+ZPAK regarding glutamatergic versus GABAergic fate in our original manuscript, we have now extended Figure 3D towards cholinergic, dopaminergic, and serotonergic neuron-specific gene expression. Also here, we did not observe any indication of differential subtype determination between NC and NC+ZPAK.

In this new analysis, we did not observe clear differences between NC and ZPAK regarding subtype-specific gene expression. However, the levels of dopaminergic and serotonergic marker genes showed a more mixed pattern at D20 between the individual cell lines, which can likely be attributed to their low levels of expression.

Thus, to further verify stable subtype commitment and to confirm the very low levels of monoaminergic neurons in our cultures, we performed immunocytochemistry using neuronal subtype-specific antibodies against GABA, vGlut1, ChAT, TH, and 5-HT to test for GABAergic, glutamatergic, cholinergic, dopaminergic, and serotonergic, respectively. These new data were added as Figure 3—figure supplement 2.

Quantifications of immunofluorescence stainings of D20 NC and NC+ZPAK iNs revealed a clear bias towards glutamatergic neurons and a smaller population of GABAergic neurons, with ~75% of NeuN+ cells staining positive for vGlut1, and ~25% of NeuN+ cells staining positive for GABA. We observed <5% of cells staining positive for markers of cholinergic, dopaminergic, or serotonergic neurons. These results are consistent with previous data (Mertens et al., 2015, Vadodaria et al., 2016), the transgenes used to reprogram in the UNA construct (Berniger et al., 2007; Heinrich et al., 2010), and with the RNA-Seq data presented in Figure 3D.

The authors mention the alternative myogenic fate (as induced by Ascl1 alone in murine fibroblasts (Treutlein et al., 2016) being more highly expressed in the NC condition as compared to NC+ZPAK. Could they confirm this via staining?

We thank the reviewer for this spot-on suggestion, and we have included immunofluorescence stainings and quantifications of the myogenic marker Myh3 on D20 iNs derived in NC and NC+ZPAK that verify the suspected loss in myogenic fate through ZPAK. These new data were added to the supplement as Figure 3—figure supplement 4.

In Figure 4C the authors show that AZ-treated cells went through significantly fewer divisions. Was cell division assessed in NC+ZAPK versus NC? In Figure 4E it looks like there are more cells in NC+ZAPK.

As described in the Materials and methods (subsection “Direct Conversion of Human Fibroblasts into iNs”), fibroblasts are pooled into high densities that on their own result in contact inhibition; only after 24h did we change to conversion media with dox, further blocking cell cycles. Thus, differences in de facto proliferation after plating are not expected to significantly influence final iN yields and densities, whereas conversion efficiency does. The observed proliferation-blocking effect of AZ, which is further accompanied by a loss of total STAT3, most likely increased conversion efficiency by promoting a postmitotic G0-like state, and thereby made the cells more amenable to adopting a neuronal identity. This explanation is consistent with previous reports of AZ960 limiting division of T cells and the known role of STAT3 in mediating cell growth and proliferation in many cell types (Yang et al., 2010; Sherry et al., 2009; Corvinus et al., 2005).

Thus, while in the representative image in Figure 4F there appear to be more cells, this observation is a direct result of more cells making it into the neuronal state with complex morphologies. Non-neuronal cells have not been traced and reconstructed and are thus not visible in this image. Instead, we feel that Figure 2D better demonstrates the described phenomenon, as it also shows similar DAPI densities, whereas more of these cells converted into βIII-tubulin-positive neurons. To further verify that not only AZ, but also ZPAK promotes a postmitotic state, we have added new CFSE experimental data that include ZPAK, and we have extended Figure 4 accordingly (see Figure 4C).

Reviewer 3 (reviewer 1 had similar questions to those described clearly below):Comparing NC and NC+ZPAK in Figure 3, it appears the authors used bulk fibroblasts undergoing iN conversion at days 5, 10, 15 and 20. However, given the boost in conversion efficiency with ZPAK (Figures 2D-H), could the fact that there are more neurons and fewer fibroblasts in the NC+ZPAK conversion condition compared to the NC condition account for the increased 'adult / defined' neuronal signature? Can the authors rule out a change in cell ratios and ability to detect small transcript changes in a sub-population rather than changes to the neuron population specifically? In order to conclude that NC+ZPAK not only improves efficiency of conversion but also generates neurons that are more 'human brain-like' the authors need to compare NC neurons to NC+ZKAK neurons directly (e.g., through transcriptional analysis of BIII-tub positive neurons, immunophenotyping of neurons to quantify specific markers that are representative of the reported signature, etc.).

This a very good point and a rather complex one to answer experimentally in its entirety. First, our functional data on morphological complexity, cytoskeletal dynamics, and mitochondrial membrane potential are all *‘specific markers’ …’ representative of the reported signature’*. These results are indicative of ‘*better neurons’*, rather than merely ‘*more neurons’*. However, we still cannot rule out that a change in cell ratios (more neurons) underlies the observed transcriptome changes at least partially.

To provide further evidence and perform a direct comparison between NC and NC+ZPK iNs, we analyzed neural activity in four separate cell lines using live Ca^2+^ imaging (the new Figure 3—figure supplement 6). Briefly, cells were transduced with lentiviral particles for synapsin dsred and GCAMP5G and underwent live calcium imaging on a spinning disk confocal (A). Panel B illustrates that overall, cells reprogrammed in NC+ZPAK had more spontaneous calcium events than those in NC alone, and that the most active cells observed were iNs reprogrammed in NC+ZPAK. When comparing the activity of all cells studied in this experiment (n=160), we observed a significant increase in the number of calcium transients produced by iNs converted in NC+ZPAK (panel C). As calcium transients have been established as a reliable readout of neural activity in vitro, this direct comparison between NC and NC+ZPAK iNs provides further evidence that ZPAK is producing a more mature and defined neuronal state (Rosenberg and Spitzer, 2011).

In order to support the authors’ conclusions about gene changes over time in Figure 3, key genes from the RNA-seq dataset should be validated by qPCR with error bars. Specifically, showing changes in key neuro, muscle and fibroblast genes over time; key neuronal genes at later time-points that corroborate the more mature phenotype implied from the Allen brain comparison in Figure 3H; glutamatergic and GABAergic genes in order to conclude that they are expressed at roughly similar levels in Figure 3D.

We have performed qPCR validation of selected key genes to verify their differential expression over time or at day 20 in NC and NC+ZPAK. The new data were added as Figure 3—figure supplement 5. SYBR green qPCR on genes highlighted in Figure 3F showed similar patterns, as identified by RNAseq, which is consistent with our observations of decreased proliferation, suppression of myogenic genes, and improved neuronal maturity and differentiation as a result of ZPAK.

Additionally, as stated above in the our first answer to reviewer #1, neuronal subtype specification in NC and NC-ZPAK was validated by extending Figure 3D towards rarer subtypes and by testing the immunofluorescence of subtype markers GABA, vGlut1, ChAT, TH, and TPH2.

The differences between NC and NC+ZPAK implied in Figures 3G and 3H seem somewhat at odds with the modest transcriptional changes reported, i.e., "NC-ZPAK iNs possess a transcriptional profile more closely resembling that of mature neurons and that the original fibroblast transcriptional program, as well as other non-neuronal directions, were substantially reduced or absent." But, the first paragraph of the subsection “ZPAK induces a defined neuronal transcription that more closely relates to the adult brain transcriptome than NC alone” indicates only 143 genes were significantly differentially expressed between NC and NC+ZPAK pooling all differences across days 5, 10, 15 and 20. How do the authors integrate these two pieces of data?

To determine DE genes, we applied p-value adjusted for multiple testing by the Benjamini-Hochberg method, and we set a false discovery rate cutoff at 0.05 (padj < 0.05), yielding 143 DE genes. However, although the list of significantly differentially expressed individual genes is not very long, when assessing populations of genes via both gene ontology analysis (Figure 3G) and RRHO (Figure 3H), we observed that classes of neuronal genes were significantly more likely to be upregulated in NC+ZPAK than NC, even when the individual genes themselves weren’t significantly differentially expressed. While this strategy is based on a significance cutoff and thus meets a comparatively high standard (that likely produces very few false-positive DE genes), we expect that, by limiting ourselves to this method, we missed many transcriptome changes. For the revised manuscript, we provide an additional strategy to challenge our view that ZPAK significantly improved the neuronal transcriptomes and that the 143 genes likely underestimated these differences.

We first selected pre-defined gene sets from databases such as the GO and KEGG and next plotted the mean and median fold change in NC versus NC-ZPAK of all genes per GO/KEGG gene set. While no statistical cutoff is provided, this type of analysis shows general expression changes of pre-defined and validated functional gene sets between NC and NC+ZPAK. For example, without applying any pre-filtering, several gene sets such as Neuron Differentiation, Neurite Development or Axon Guidance (GO terms) showed an upregulation in NC+ZPAK both when assessing the mean and median expression changes. These data were added as Figure 3—figure supplement 3.

The authors should also include the number of differentially expressed genes for each time-point and the full list of 143 DEGs, as supplementary figures. Since this is a small/important gene set for this study, it should be readily accessible.

We agree that the list of significantly differentially expressed genes at each time point should be included in an easily accessible manner. We thus included a list of all DE genes in a table as part of the supplementary information as Figure 3—source data 1.

Does the transcript data support the mechanistic data in Figures 4-5? Is there transcriptional evidence to support the signaling pathways induced by ZPAK over the course of conversion? (if so, again, specific gene changes should be validated from the RNA-seq). Upon activation of these signaling pathways, and dramatic differences in neurite complexity shown in Figures 4E-F, again one might expect to see more differentially expressed genes between NC and NC-ZPAK.

As the reviewer suggests, there is encouraging but modest transcriptional evidence supporting the mechanistic data in Figure 4, with the Log2 fold change trending in the direction of the changes reported, but these differences in transcript levels aren’t significant when controlled for multiple observations as is necessary with RNA-Seq analysis. We thus agree that the list of significantly differentially expressed genes between NC and NC+ZPAK, with 143 genes at day 20, appears relatively small at first sight. As discussed above, however, we assume that the strict significance cutoff, the sampling from only three fibroblast donors (which for timeline RNA-Seq is already quite work- and cost-intensive), and the fact that bulk RNA was assessed might produce many false-negative genes. For the revised manuscript, we attempted to broaden the view on the transcriptome changes. For example, we show that gene sets that directly relate to the results presented in Figure 4E-F, such as Axon Guidance and Neurite Development, are generally upregulated in ZPAK (see our fourth answer to reviewer #3). In addition to the differences in neurite complexity, we have also observed significantly increased neural activity as assessed by live cell calcium imaging (see our first answer to reviewer #3). Thus, we reason that the accumulated impact of transcriptional bias towards neuronal fate induced by ZPAK results in iNs with a more defined and mature neuronal phenotype.

3) Usefulness of the new protocol.The authors are correct to point out that efficiency matters for investigators trying to produce cultures enriched in neurons and for those trying to make large numbers of neurons for proteomic or screening applications. However, by way of comparison, direct differentiation from iPSCs involves generating neurons from pluripotent cells that can be grown in virtually unlimited numbers. In the case of the current method, the numbers of neurons that can be produced from fibroblasts is the product of the number of fibroblasts that can be generated from each patient sample X the efficiency of differentiation. The authors have shown that the efficiency of their method is better, but how well does the method work in fibroblasts that have been passaged sufficiently to give rise to the billions of neurons that may be required? The lines used in this work were already passaged >10 times, which is promising. However, the authors should demonstrate the process will continue to work on these fibroblast lines, especially those derived from older donors, that are passaged sufficiently to produce the requisite large numbers of neurons.

The ability to make large numbers of iNs is indeed important for future application of this technology, and the reviewer is correct to highlight this significant goal of the improved protocol. As the fibroblasts are the only cells in this protocol that can be expanded and passaged to produce large numbers of cells, the iN approach from non-transformed fibroblasts will never yield the cell numbers that iPSC can deliver. Here, the limitation is the fibroblasts and not the conversion efficiency, and the scope of this manuscript is improving iN conversion, not extending fibroblast passages and proliferation speed, which probably results in detrimental effects on the beneficial aspects of the iN technology, such as age preservation.

However, to get the best from the fibroblasts as a cell source, we sought to verify that iNs can be obtained from highly passaged (>30) fibroblasts. To this end, we produced 4 lines with >30 passages to test for direct conversion with UNA and NC+ZPAK. We verified that these high passage cells were approaching replicative senescence by SA-B-Gal staining. First, we observed that the overall efficiency was reduced in both conditions when compared to early passages, which was expected based reports of senescence-limiting iN conversion (Sun et al., Nat Commun 2014). However, we found that NC+ZPAK still produced a significantly increased yield of iNs compared to NC alone also from late passage fibroblasts. We have included these data in Figure 5—figure supplement 1.